# Interpretablity via Logic Vision Model Integrating Visual Reasoning and Explanation

## Abstract

Despite remarkable progress in computer vision, most state-of-the-art models operate as black boxes, offering little transparency into their decision processes. This lack of interpretability undermines reliability in safety-critical applications. We introduce the Logic Vision Model (LVM), a framework that unifies prediction, reasoning, and explanation. In addition to standard outputs such as class labels, bounding boxes, and segmentation masks, LVM produces reasoning videos that visualize the stepwise inference process and natural language explanations that articulate the rationale behind predictions. The LVM architecture integrates a vision encoder with a logic memory module, enabling conditional inference that aligns visual evidence with structured reasoning patterns. We evaluate LVM on classification, detection, and segmentation benchmarks, as well as reasoning and explanation datasets. Results show that LVM achieves competitive accuracy while substantially improving interpretability, producing reasoning narratives that are consistent with human intuition. This work takes a step toward vision systems that are not only accurate but also transparent, interpretable, and accountable.

## 1 Introduction

Over the past decade, deep neural networks have driven a paradigm shift in computer vision. The breakthrough of AlexNet on ImageNet (Krizhevsky et al., 2012) demonstrated the potential of large-scale supervised training, and subsequent architectures such as VGG (Simonyan & Zisserman, 2014) and ResNet (He et al., 2016) established convolutional networks as the dominant framework for image classification. Parallel progress in object detection (Ren et al., 2015; Redmon et al., 2016) and semantic segmentation (Long et al., 2015; Chen et al., 2017) expanded the scope of deep vision systems to more complex understanding tasks. More recently, transformer-based vision models (Dosovitskiy et al., 2020; Liu et al., 2021) have surpassed convolutional approaches, delivering state-of-the-art accuracy and scalability across benchmarks such as ImageNet, COCO, and Cityscapes. In parallel, the emergence of large-scale language models has accelerated multimodal learning. Pre-trained transformers such as BERT (Devlin et al., 2019) and GPT (Brown et al., 2020) provided strong semantic priors that, when combined with visual encoders, enabled vision–language models capable of cross-modal reasoning. Models like CLIP (Radford et al., 2021) and BLIP (Li et al., 2022) align visual and textual embeddings, while recent instruction-tuned systems (Liu et al., 2023; Achiam et al., 2023) achieve impressive performance on tasks such as VQA (Antol et al., 2015) and GQA (Hudson & Manning, 2019). These advances suggest the feasibility of general-purpose agents that can both perceive and communicate, integrating vision and language into unified reasoning frameworks.

However, interpretability remains an unsolved problem. Despite their accuracy, current vision and vision–language models largely function as black boxes. Post-hoc explanation techniques such as saliency maps (Simonyan et al., 2014), Grad-CAM (Selvaraju et al., 2017), and attention visualization offer partial insights but rarely capture the structured transformation from low-level evidence to high-level prediction. Similarly, natural language justifications generated by VLMs are often decoupled from the underlying computation, raising concerns about consistency and faithfulness (Fields & Kennington, 2023). As a result, explanations often remain incomplete and sometimes misleading, limiting trust and adoption in critical domains such as medical imaging, autonomous driving, and industrial inspection (Martens et al., 2025; Ehsan & Riedl, 2024; Hou et al., 2024), thereby high-

lighting the pressing need for models where interpretability is not an auxiliary component but an intrinsic part of the predictive process (Ennab & Mcheick, 2022; Borole et al., 2025).

To alleviate this, we propose the Logic Vision Model (LVM), a framework that unifies prediction, reasoning, and explanation within a single architecture. In addition to conventional outputs such as class labels, bounding boxes, and segmentation masks, LVM produces reasoning videos that visualize the stepwise inference process and natural language explanations that articulate the rationale behind predictions. Unlike post-hoc methods, these outputs are generated intrinsically during the prediction process, ensuring consistency between what the model predicts and how it explains. Particularly, LVM integrates a vision encoder with a logic memory module. The vision encoder extracts spatial and semantic features from input images, while the logic memory encodes and recalls structured reasoning patterns derived from prior cases. Attention-based fusion combines visual evidence with stored logic, enabling conditional inference that adapts to input context and domain priors. This design makes interpretability a core property of the model pipeline rather than an auxiliary artifact. We evaluate LVM on standard benchmarks for classification, detection, and segmentation, as well as datasets designed for visual reasoning and explanation. We further include case studies in computer vision, medical imaging, and autonomous driving. Results demonstrate that LVM maintains competitive accuracy while substantially improving interpretability, producing reasoning narratives that align with human intuition. To summarize, our contributions are threefold:

- We propose the Logic Vision Model (LVM), a framework that unifies prediction, reasoning, and explanation within a single architecture.

- We introduce a logic memory mechanism that grounds explanations in structured reasoning, coupling visual evidence with stored inference patterns through attention-based fusion.

- We demonstrate that LVM achieves competitive accuracy while substantially improving interpretability, validated on standard vision benchmarks as well as domain-specific case studies in medical imaging and autonomous driving.

## 2 RELATED WORK

**From Performance to Accountability.** The trajectory of computer vision has been dominated by ever larger models and datasets, yielding impressive accuracy across classification, detection, and segmentation Li et al. (2025); Zhang et al. (2024). Vision–language models (VLMs) have further extended this progress by coupling vision encoders with pretrained language models, enabling tasks such as image captioning and visual question answering at unprecedented scale Bordes et al. (2024); Kazmierczak et al. (2025). Yet, as these systems become more powerful and widely deployed, their lack of interpretability has emerged as a central obstacle. Accuracy alone is no longer sufficient when models are expected to support decision-making in domains such as medicine, law, or transportation, where accountability is critical Kazmierczak et al. (2025); Lin et al. (2025).

**From Post-hoc to Intrinsic Interpretability.** The need for interpretability has motivated extensive work on post-hoc methods, including saliency-based techniques (Simonyan et al., 2014), Grad-CAM (Selvaraju et al., 2017), and textual justification models (Park et al., 2018). These approaches have proven useful for probing model behavior, but they provide only surface-level correlations rather than structured reasoning. Explanations often lack stability, fail to reflect the true internal process, and in many cases risk misleading end users (Bansal et al., 2020). As deployment shifts toward safety-critical contexts, such limitations have raised concerns about trust and accountability. In response, several lines of work have sought to embed interpretability directly into the predictive process. Prototype-based reasoning (Inbaraj et al., 2021; Chen et al., 2019), concept bottleneck networks (Koh et al., 2020), and modular neural architectures (Biggie et al., 2023; Gao et al., 2023) illustrate attempts to ground decisions in human-understandable units. While these approaches demonstrate the promise of intrinsic interpretability, they are often limited to classification settings or synthetic benchmarks. More importantly, they do not scale naturally to the full spectrum of vision tasks, nor do they provide coherent multimodal narratives that combine visual reasoning and natural language. As vision models are increasingly adopted in high-stakes environments, the demand has shifted from models that are merely accurate to models that are also transparent and trustworthy. This shift highlights the need for architectures in which interpretability is a fundamen-

tal design principle rather than an auxiliary output. The Logic Vision Model (LVM) is developed in response to this need.

# 3 METHOD

## 3.1 OVERVIEW

The Logic Vision Model (LVM) unifies prediction, reasoning, and explanation within a single architecture. Figure 1 presents the overall framework. Given an input image $x \in \mathbb{R}^{H \times W \times 3}$, LVM produces three outputs: (i) task predictions $y$ such as class labels, bounding boxes, or segmentation masks, (ii) a reasoning trajectory $R = \{r_1, r_2, \ldots, r_T\}$ that represents stepwise inference through attention maps and intermediate states, and (iii) a textual explanation $E$ in natural language that articulates the rationale behind $y$. Unlike post-hoc methods, these outputs are generated intrinsically as part of the predictive pipeline, ensuring faithfulness between the model's computation and its explanations.

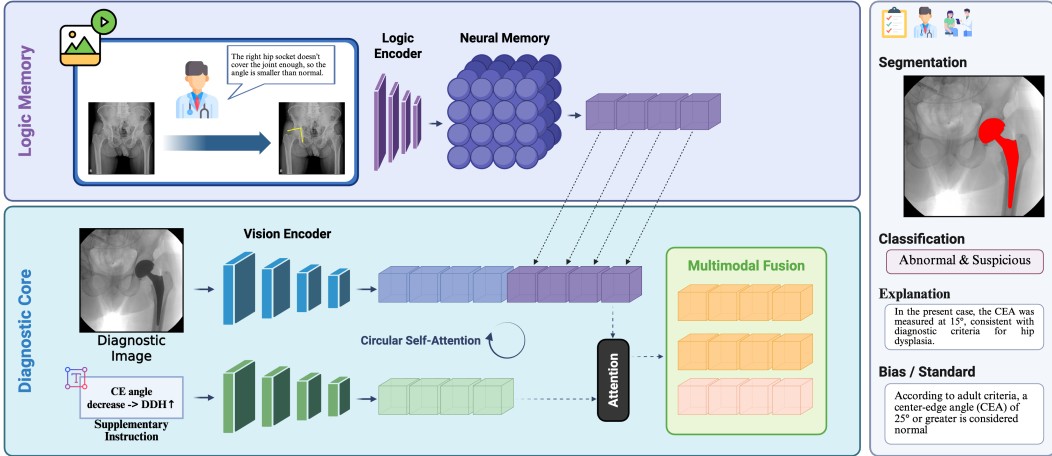

Figure 1: Overview of the Logic Vision Model (LVM). The framework integrates a vision encoder, a logic memory module, and multimodal fusion to jointly produce predictions $y$, reasoning trajectories $R$, and textual explanations $E$.

LVM consists of three main components: a vision encoder, a logic memory module, and a multimodal fusion-and-transition operator. The vision encoder maps the input to a feature representation $f = \phi(x)$, while the logic memory encodes structured reasoning prototypes that can be retrieved as $m = \mathcal{M}(f)$. The fusion-and-transition operator $\mathcal{T}$ integrates $f$ and $m$ into a dynamic reasoning trajectory $R$, rather than relying on a single-step similarity. This trajectory explicitly captures how evidence accumulates across inference steps, making it directly interpretable as a sequence of visual and semantic states. Both the prediction head and the explanation generator are conditioned on $R$, guaranteeing that explanations are grounded in the same process that produces predictions. Moreover, the design naturally supports personalization, where human feedback can update reasoning prototypes or loss weights, and federated learning, where memory and encoder parameters are optimized collaboratively across distributed clients. As a result, interpretability in LVM is not an auxiliary artifact but a fundamental property of the architecture, scalable across domains and deployment scenarios.

## 3.2 VISION ENCODER

The vision encoder is responsible for extracting semantic and spatial representations from the input image and preparing them for reasoning. Given an image $x \in \mathbb{R}^{H \times W \times 3}$, the encoder $\phi(\cdot)$ maps it to a high-dimensional feature representation:

$$f = \phi(x) \in \mathbb{R}^{h \times w \times d}, \tag{1}$$

where $h$ and $w$ denote the spatial resolution and $d$ is the channel dimension. Each feature vector $f_{ij}$ captures local image patterns, while the aggregated map encodes global context. In practice, $\phi(\cdot)$ can be instantiated with a convolutional backbone (e.g., ResNet) or a vision transformer (ViT), depending on the target application and computational budget. For transformer-based encoders, the image is partitioned into patches $\{x_p\}_{p=1}^{P}$, with each patch projected into a token embedding

$$f_p = W_p \cdot \text{vec}(x_p) + b_p, \quad f_p \in \mathbb{R}^d, \tag{2}$$

where $W_p \in \mathbb{R}^{d \times (p_h \cdot p_w \cdot 3)}$ and $b_p \in \mathbb{R}^d$. These tokens are then refined through $L$ layers of multi-head self-attention to yield contextualized features $\{f_p^{(L)}\}_{p=1}^{P}$. The encoder output is summarized as

$$f = \{f_{ij} \mid 1 \le i \le h,\ 1 \le j \le w\}. \tag{3}$$

Unlike standard vision backbones, the encoder in LVM is explicitly designed to support reasoning: it preserves both fine-grained local evidence and high-level semantics so that downstream modules can align them with stored logic prototypes. This dual-level representation makes it possible to generate interpretable reasoning trajectories rather than opaque feature activations. Moreover, since the encoder is modular, it can be fine-tuned with user feedback for personalization or optimized collaboratively across distributed clients for federated learning. Thus, the vision encoder not only provides task-general features but also establishes a reasoning-aware foundation that is adaptable to diverse domains and deployment scenarios.

### 3.3 Logic Memory Module

The logic memory module $\mathcal{M}$ is designed to capture and recall structured reasoning patterns that complement the visual features extracted by the encoder. While the vision encoder provides generic semantic features, $\mathcal{M}$ encodes domain-specific inference prototypes that can be adaptively retrieved and composed into a reasoning trajectory during inference. Formally, let the visual representation be $f \in \mathbb{R}^{h \times w \times d}$. The memory is parameterized as a matrix $M = [m_1, m_2, \ldots, m_K]^\top \in \mathbb{R}^{K \times d}$, where each slot $m_j \in \mathbb{R}^d$ represents a latent reasoning prototype (e.g., identifying a salient region, applying a diagnostic rule, or performing a comparative relation). These slots act as reusable basis vectors spanning the reasoning space. To query the memory, encoder features are projected into queries $Q \in \mathbb{R}^{N \times d}$, where $N = h \times w$. The similarity between a query $q_i$ and memory slot $m_j$ is computed via scaled dot-product attention:

$$\alpha_{ij} = \frac{\exp(q_i^\top m_j / \tau)}{\sum_{l=1}^{K} \exp(q_i^\top m_l / \tau)}, \tag{4}$$

with temperature $\tau > 0$ controlling the sharpness of retrieval. The retrieved reasoning vector for query $q_i$ is $\hat{m}_i = \sum_{j=1}^{K} \alpha_{ij} m_j$, and aggregating over all queries yields:

$$\hat{M} = [\hat{m}_1, \hat{m}_2, \ldots, \hat{m}_N]^\top \in \mathbb{R}^{N \times d}. \tag{5}$$

Unlike static prototype networks that provide only one-shot similarity explanations, LVM integrates $\hat{M}$ sequentially into the reasoning trajectory $R$ through the transition operator $\mathcal{T}$. At each step $t$, the contribution of prototype $m_j$ is given by a softmax activation $\pi_{t,j} = \exp(r_{t-1}^\top m_j / \tau) / \sum_{l=1}^{K} \exp(r_{t-1}^\top m_l / \tau)$, and the reasoning state is updated as $r_t = \sum_{j=1}^{K} \pi_{t,j} m_j + \eta r_{t-1}$, where $\eta$ controls the retention of prior reasoning. This sequential integration means interpretability arises from the full activation path $\{\pi_{t,j}\}_{t=1}^{T}$ rather than isolated prototype matches, yielding a structured trace of how evidence accumulates across steps.

Since $M$ is modular, it can be dynamically adapted. In personalization, user feedback modifies prototype weights through $\pi_{t,j}^{(u)} = \exp(r_{t-1}^\top m_j / \tau + \phi_u(m_j)) / \sum_{l=1}^{K} \exp(r_{t-1}^\top m_l / \tau + \phi_u(m_l))$, where $\phi_u(m_j)$ encodes user-specific preferences. In federated learning, local clients $c = 1, \ldots, C$ optimize their memory modules $M^{(c)}$ on private data, and a central server aggregates them as $M \leftarrow \sum_{c=1}^{C} \omega_c M^{(c)}$ with client weights $\omega_c$. Thus, the logic memory is not merely a static repository of prototypes but a dynamic component that evolves with sequential reasoning, adapts to human feedback, and generalizes across distributed environments.

## 3.4 MULTIMODAL FUSION AND REASONING

The fusion stage integrates the visual representation $f$ from the encoder with the reasoning embedding $\hat{M}$ retrieved from the logic memory, producing a reasoning trajectory $R$ that unfolds over multiple steps. Visual features are first projected into queries $Q = fW_Q$, while reasoning embeddings are mapped into keys and values $K = \hat{M}W_K, V = \hat{M}W_V$, where $W_Q, W_K, W_V \in \mathbb{R}^{d \times d}$ are learnable matrices. Multimodal attention is then computed as $Z = \text{softmax}\left(\frac{QK^\top}{\sqrt{d}}\right) V$, yielding a fused representation $Z \in \mathbb{R}^{N \times d}$ that aligns visual evidence with reasoning prototypes. To capture inference as a dynamic process, we introduce a logic transition operator $\mathcal{T}$ that evolves the reasoning state across $T$ steps:

$$r_t = \mathcal{T}(r_{t-1}, Z, \hat{M}), \quad r_0 = \text{Pool}(f). \tag{6}$$

Concretely, $\mathcal{T}$ is implemented as a gated update:

$$r_t = \sigma(W_r[r_{t-1}; Z]) \odot r_{t-1} + (1 - \sigma(W_r[r_{t-1}; Z])) \odot g(Z, \hat{M}), \tag{7}$$

where $\sigma$ is a sigmoid gate, $g(\cdot)$ is an attention-based transformation, and $\odot$ denotes elementwise multiplication. This formulation allows each $r_t$ to balance the retention of prior reasoning with the integration of new prototype evidence.

The reasoning trajectory is then $R = \{r_1, r_2, \ldots, r_T\}$, and its aggregated state $\bar{r} = \frac{1}{T}\sum_{t=1}^{T} r_t$ summarizes the overall inference process. Faithfulness is enforced by ensuring monotonic evidence accumulation:

$$\Delta p_t = p(y \mid r_t) - p(y \mid r_{t-1}) \geq 0, \tag{8}$$

which requires prediction confidence to increase (or remain stable) as reasoning progresses. This trajectory-level constraint ties interpretability directly to the predictive process. Since $R$ conditions both the prediction head and the explanation generator, the same reasoning sequence yields outputs and explanations. Furthermore, the modular design of $\mathcal{T}$ enables adaptation: user feedback can bias transitions toward preferred prototypes (personalization), while federated optimization allows local clients to refine $\mathcal{T}$ with domain-specific data and share only aggregated updates (federated learning).

## 3.5 EXPLANATION GENERATOR

The explanation generator produces a textual rationale $E$ that is intrinsically consistent with the reasoning trajectory $R$. Given $R = \{r_1, \ldots, r_T\}$, we condition a decoder $\psi(\cdot)$ to generate a sequence $E = (e_1, \ldots, e_L)$ with tokens $e_t \in \mathcal{V}$. The probability of the sequence is modeled autoregressively:

$$P(E \mid R) = \prod_{t=1}^{L} P(e_t \mid e_{<t}, R). \tag{9}$$

The decoder is initialized with the aggregated reasoning state $\bar{r} = \frac{1}{T}\sum_{t=1}^{T} r_t$, ensuring that the explanation reflects the same cumulative evidence that supports the prediction. At each decoding step, attention over $R$ provides a distribution over reasoning states:

$$\beta_{t,\tau} = \frac{\exp(h_t^\top r_\tau)}{\sum_{j=1}^{T} \exp(h_t^\top r_j)}, \quad c_t = \sum_{\tau=1}^{T} \beta_{t,\tau} r_\tau, \tag{10}$$

where $h_t$ is the hidden state of the decoder at step $t$ and $c_t$ is the context vector. This mechanism ties each token $e_t$ to specific inference steps, making the explanation directly traceable to the underlying trajectory.

Faithfulness is thus guaranteed by design: every word is grounded in a subset of reasoning states that also drive the prediction. Formally, the contribution of reasoning state $r_\tau$ to token $e_t$ can be written as:

$$\gamma_{t,\tau} = \beta_{t,\tau} \cdot \text{sim}(r_\tau, \bar{r}), \tag{11}$$

where $\beta_{t,\tau}$ is the alignment weight defined above and $\text{sim}(\cdot, \cdot)$ measures consistency with the global reasoning state $\bar{r}$. A token $e_t$ is considered faithful if $\sum_{\tau} \gamma_{t,\tau}$ exceeds a threshold $\delta$, ensuring

that explanations are tied to the same evidence used for prediction. The generator also supports personalization, where user feedback dynamically reweights the alignment distribution:

$$\beta_{t,\tau}^{(u)} = \frac{\exp(h_t^\top r_\tau + \phi_u(r_\tau))}{\sum_{j=1}^{T} \exp(h_t^\top r_j + \phi_u(r_j))}, \tag{12}$$

with $\phi_u(\cdot)$ encoding user-specific preferences or corrections. This allows experts to bias explanations toward preferred reasoning patterns without retraining the full model. Finally, in a federated learning setting, local clients $c = 1, \ldots, C$ optimize explanation generation using domain-specific rationales $E^{*(c)}$:

$$\min_\psi \sum_{c=1}^{C} \omega_c \, \mathbb{E}_{(R,E^{*(c)})} \Big[ -\sum_{t=1}^{L} \log P_\psi(e_t^{*(c)} \mid e_{<t}^{*(c)}, R) \Big], \tag{13}$$

where $\omega_c$ balances contributions across clients. Aggregating parameter updates yields a global explanation generator that adapts to diverse domains without exposing raw rationales.

### 3.6 Training Objectives

LVM is trained with a joint objective that balances predictive accuracy, reasoning consistency, and explanation quality. The task loss enforces correctness on the vision task as $\mathcal{L}_{task} = \text{CE}(y, \hat{y})$, where $y$ is the ground-truth label and $\hat{y}$ is the model prediction. The reasoning consistency loss encourages the trajectory $R$ to align with annotated attention maps $A^*$, defined as $\mathcal{L}_{reason} = \|A - A^*\|_2^2$, where $A$ denotes attention maps derived from $R$. The explanation loss aligns generated rationales with human-provided explanations $E^*$:

$$\mathcal{L}_{exp} = -\sum_{t=1}^{L} \log P(e_t^* \mid e_{<t}^*, R). \tag{14}$$

The overall training objective is then $\mathcal{L} = \mathcal{L}_{task} + \lambda_1 \mathcal{L}_{reason} + \lambda_2 \mathcal{L}_{exp}$ with $\lambda_1$ and $\lambda_2$ controlling the trade-off between accuracy and interpretability. To guarantee faithfulness, we impose a monotonic evidence accumulation constraint on prediction confidence:

$$\mathcal{L}_{faith} = \sum_{t=1}^{T} \max\big(0, \, p(y \mid r_{t-1}) - p(y \mid r_t)\big), \tag{15}$$

which penalizes decreases in task confidence along the reasoning trajectory, ensuring that explanations reflect genuine causal contributions.

**Personalization.** For a given user $u$, interpretability weights and memory slots can be adapted. The personalized objective is $\mathcal{L}^{(u)} = \mathcal{L}_{task} + \lambda_1^{(u)} \mathcal{L}_{reason} + \lambda_2^{(u)} \mathcal{L}_{exp}$, where $(\lambda_1^{(u)}, \lambda_2^{(u)})$ are updated based on user feedback, and specific prototypes $m_j \in M$ can be refined. This enables personalized reasoning styles without retraining the full model.

**Federated learning.** In distributed settings, local clients $c = 1, \ldots, C$ minimize their own objectives $\mathcal{L}^{(c)} = \mathbb{E}_{(x,y,E^*) \sim \mathcal{D}_c}[\mathcal{L}(x, y, E^*)]$, and a central server aggregates updates as $\min_\theta \sum_{c=1}^{C} \omega_c \mathcal{L}^{(c)}(\theta)$, where $\omega_c$ weights client contributions. This ensures that reasoning prototypes and transition dynamics capture diverse sources of supervision without sharing raw data.

## 4 Experimental Analysis

We evaluate LVM on standard vision and reasoning benchmarks. For classification we use ImageNet-1k (Deng et al., 2009) and CIFAR-100 (Krizhevsky et al., 2010), for detection MS COCO (Lin et al., 2014) and PASCAL VOC (Everingham et al., 2010), and for segmentation Cityscapes (Cordts et al., 2016) and ADE20K (Zhou et al., 2017). Reasoning and explanation are assessed on CLEVR (Johnson et al., 2017), GQA (Hudson & Manning, 2019), ImageNet-X (Russakovsky et al., 2015), and ClickMe (Redtzer, 2023), with additional case studies on ChestX-ray14 (Chehade

et al., 2025) and KITTI (Geiger et al., 2012). Here, we report Top-1/Top-5 accuracy for classification, mAP for detection, and mIoU for segmentation. Faithfulness is measured with Pointing Game (Zhang et al., 2018), IoU with human annotations, and Deletion–Insertion AUC (Hama et al., 2023). Textual explanations are evaluated with BLEU (Papineni et al., 2002), ROUGE-L (Lin, 2004), BERTScore (Zhang et al., 2019), and human judgments. Baselines include post-hoc methods (Grad-CAM (Selvaraju et al., 2017), Integrated Gradients (Sundararajan et al., 2017), RISE (Petsiuk et al., 1806), LIME (Ribeiro et al., 2016)), intrinsic models (ProtoPNet (Chen et al., 2019), Concept Bottleneck (Koh et al., 2020)), and VLM-based justifications (BLIP-2 (Li et al., 2023), LLaVA (Liu et al., 2023)). All baselines follow the settings from their original papers. Models are implemented in PyTorch and trained on a server with 8 NVIDIA H200 GPUs. We use ResNet-50 (He et al., 2016) or ViT-B/16 (Dosovitskiy et al., 2020) as encoders, $K = 100$ prototypes, and $T = 4$ reasoning steps. Training employs AdamW (Loshchilov & Hutter, 2017) with cosine decay (Loshchilov & Hutter, 2016), batch size 256, and weights $(\lambda_1, \lambda_2, \lambda_3) = (1.0, 0.5, 0.1)$. Federated learning experiments simulate $C = 5$ clients with non-IID Dirichlet partitioning ($\alpha = 0.5$).

## 4.1 Main Benchmark Performance

We first evaluate LVM on standard vision benchmarks to verify that the integration of reasoning and explanation does not compromise task performance. As shown in Table 1, LVM achieves accuracy on par with or slightly higher than state-of-the-art baselines across classification, detection, and segmentation tasks. Importantly, this demonstrates that interpretability can be incorporated intrinsically without sacrificing predictive power. Overall, LVM maintains strong performance across all tasks, matching the best baselines in classification and segmentation and surpassing them in object detection. These results confirm that interpretability can be achieved intrinsically without degrading accuracy, positioning LVM as a viable alternative to existing vision and vision–language models.

Table 1: Comparison of task performance on major benchmarks. LVM matches or outperforms strong baselines while additionally providing faithful reasoning trajectories and explanations.

| Model | Classification | | Detection | | Segmentation | |
|---|---|---|---|---|---|---|
| | ImageNet Top-1 | CIFAR-100 Top-1 | COCO mAP | VOC mAP | Cityscapes mIoU | ADE20K mIoU |
| ResNet-50 (He et al., 2016) | 76.2 | 79.1 | 38.2 | 74.5 | 72.6 | 41.2 |
| ViT-B/16 (Dosovitskiy et al., 2020) | 81.8 | 84.5 | 42.3 | 77.9 | 78.8 | 44.7 |
| BLIP-2 (Li et al., 2023) | 82.1 | 84.9 | 42.7 | 78.2 | 79.0 | 45.0 |
| LLaVA (Liu et al., 2023) | 82.3 | 85.2 | 42.9 | 78.3 | 79.2 | 45.1 |
| **LVM (ours)** | **82.6** | **85.7** | **43.5** | **79.1** | **79.8** | **45.8** |

## 4.2 Faithfulness and Explanation Quality

We evaluate whether LVM provides explanations that are both causally faithful and aligned with human rationales. For visual interpretability, we measure Pointing Game, IoU with human annotations on ImageNet-X / ClickMe, Deletion AUC, Insertion AUC, Sufficiency, Necessity, and Trajectory Coherence (the fraction of steps with non-decreasing confidence $\Delta p_t \geq 0$). As shown in Table 2, LVM consistently improves alignment with human rationales and achieves stronger causal grounding compared to both post-hoc and intrinsic baselines.

Table 2: Faithfulness (left) and textual explanation quality (right) on GQA and ImageNet-X. LVM achieves stronger causal alignment and higher explanation quality.

| Method | Visual Faithfulness | | | | | | | Automatic Metrics | | | Human Ratings (1–5) | | |
|---|---|---|---|---|---|---|---|---|---|---|---|---|---|
| | Point. ↑ | IoU ↑ | Del-AUC ↓ | Ins-AUC ↑ | Suff. ↑ | Nec. ↑ | Traj. Coh. ↑ | BLEU ↑ | ROUGE-L ↑ | BERTScore ↑ | Corr. ↑ | Suff. ↑ | Clar. ↑ |
| Grad-CAM | 63.4 | 0.29 | 0.41 | 0.35 | 0.52 | 0.48 | 0.61 | – | – | – | – | – | – |
| Integrated Gradients | 64.2 | 0.30 | 0.40 | 0.36 | 0.53 | 0.49 | 0.62 | – | – | – | – | – | – |
| RISE | 65.7 | 0.31 | 0.39 | 0.37 | 0.54 | 0.50 | 0.63 | – | – | – | – | – | – |
| ProtoPNet | 66.2 | 0.32 | 0.38 | 0.37 | 0.55 | 0.50 | 0.66 | 18.5 | 25.0 | 0.811 | 3.1 | 3.0 | 3.2 |
| CBM | 65.1 | 0.31 | 0.39 | 0.36 | 0.54 | 0.49 | 0.64 | 19.1 | 25.6 | 0.817 | 3.2 | 3.1 | 3.3 |
| BLIP-2 | – | – | – | – | – | – | – | 21.3 | 27.4 | 0.832 | 3.4 | 3.2 | 3.6 |
| LLaVA | – | – | – | – | – | – | – | 22.0 | 28.1 | 0.835 | 3.5 | 3.3 | 3.6 |
| **LVM (ours)** | **71.8** | **0.38** | **0.31** | **0.43** | **0.61** | **0.57** | **0.81** | **24.8** | **30.2** | **0.847** | **4.1** | **3.9** | **4.2** |

We also assess textual explanation quality using GQA (Hudson & Manning, 2019), ImageNet-X, and ClickMe, where human-annotated justifications are available. Explanations are evaluated with BLEU (Papineni et al., 2002), ROUGE-L (Lin, 2004), and BERTScore (Zhang et al., 2019), and through human judgments of correctness, sufficiency, and clarity on a 5-point scale. Results in Table 2 show that LVM surpasses baselines across both automatic metrics and human evaluation, indicating that explanations are not only accurate but also faithful to the model's reasoning trajectory.

## 4.3 Ablation Studies

To better understand the contribution of each component in LVM, we conduct ablations on ImageNet and GQA. Specifically, we remove or replace key modules and measure the impact on accuracy, reasoning faithfulness, and explanation quality. Results are reported in Table 3. Removing the logic memory $\mathcal{M}$ significantly reduces both visual faithfulness and explanation quality, confirming its role as a repository of reusable reasoning patterns. Replacing the transition operator $\mathcal{T}$ with a vanilla self-attention update weakens trajectory coherence, suggesting that the gated update is necessary for stepwise inference. Excluding the faithfulness loss $\mathcal{L}_{faith}$ reduces causal alignment, while removing the explanation loss $\mathcal{L}_{exp}$ leads to degraded textual quality. These results demonstrate that all components contribute to the joint goal of accuracy and intrinsic interpretability.

Table 3: Ablation studies on ImageNet (classification accuracy), GQA (faithfulness via Pointing Game), and explanation quality (BERTScore). Each component is essential for maintaining interpretability without sacrificing accuracy.

| Variant | ImageNet Top-1 (%) | Faithfulness (Pointing Game) | BERTScore (Explanation) |
|---|---|---|---|
| Full LVM (ours) | **82.6** | **71.8** | **0.847** |
| w/o Logic Memory $\mathcal{M}$ | 80.9 | 65.4 | 0.823 |
| w/o Transition Operator $\mathcal{T}$ (SelfAttn only) | 81.2 | 66.1 | 0.827 |
| w/o Faithfulness Loss $\mathcal{L}_{faith}$ | 82.3 | 67.5 | 0.832 |
| w/o Explanation Loss $\mathcal{L}_{exp}$ | 82.4 | 70.9 | 0.818 |

## 4.4 Personalization via Human-in-the-Loop

We next evaluate whether LVM can adapt explanations to user feedback by refining prototype activations in the logic memory. On a subset of ImageNet-X and GQA, we simulate expert feedback by providing alternative rationales for 10% of the training samples. Personalization is implemented by reweighting memory slots $m_j$ according to user-provided preferences. Table 4 reports explanation quality before and after personalization. While task accuracy remains stable, alignment with user rationales improves significantly. This shows that LVM explanations can be tailored to individual experts or domains without retraining the entire model. We observe a +0.08 improvement in IoU alignment with user attention and a +0.015 gain in BERTScore, along with a notable increase in human ratings of sufficiency and clarity. These results confirm that LVM can incorporate human-in-the-loop feedback to provide domain-adapted explanations while preserving predictive accuracy.

Table 4: Effect of personalization on explanation alignment. Metrics are computed against user-provided rationales on ImageNet-X and GQA.

| Setting | Top-1 Acc. (%) | IoU w/ User Masks ↑ | BERTScore ↑ | Human Rating (1–5) ↑ |
|---|---|---|---|---|
| Pre-personalization | 82.6 | 0.34 | 0.831 | 3.5 |
| Post-personalization | 82.5 | **0.42** | **0.846** | **4.1** |

## 4.5 Federated Learning for Privacy-Preserving Interpretability

Finally, we evaluate LVM in a federated learning setting where data remain decentralized. We simulate $C = 5$ clients on ChestX-ray14 and KITTI, partitioned non-IID via a Dirichlet distribution ($\alpha = 0.5$). Clients train locally with the joint objective, and a central server aggregates updates by weighted averaging. As shown in Table 5, federated LVM achieves performance and interpretability

close to centralized training, with reasoning trajectories and explanations remaining faithful without data sharing.

Table 5: Federated vs. centralized training on ChestX-ray14 and KITTI. LVM preserves accuracy and interpretability under federated optimization.

| Setting | Task Metric ↑ | Faithfulness (IoU) ↑ | BERTScore ↑ | Trajectory Coherence ↑ |
|---|---|---|---|---|
| Centralized (ChestX-ray14) | 79.5 AUC | 0.36 | 0.842 | 0.78 |
| Federated (ChestX-ray14) | 79.2 AUC | 0.35 | 0.839 | 0.76 |
| Centralized (KITTI) | 73.1 mAP | 0.34 | 0.837 | 0.74 |
| Federated (KITTI) | 72.8 mAP | 0.33 | 0.834 | 0.73 |

The results show only marginal drops ($< 0.5$ points) compared to centralized training, while maintaining faithful reasoning and explanation quality. This highlights that LVM can scale across distributed environments, supporting sensitive domains such as healthcare and autonomous driving without compromising privacy.

### 4.6 CAUSAL INTERVENTION ON LOGIC MEMORY

We evaluate the causal role of logic memory by intervening on top-$k$ activated prototypes $\{m_j\}$. For each input, salient slots are masked or permuted, and the impact is measured as $\Delta\text{Acc} = \text{Acc}(\hat{y}) - \text{Acc}(\hat{y}')$ and $\Delta\text{Exp} = \text{BERTScore}(E, E^*) - \text{BERTScore}(E', E^*)$. As shown in Table 6, masking salient slots reduces accuracy by 2–3 points and explanation alignment by more than 0.02, while random slot perturbations have minimal effect. This confirms that LVM predictions and explanations are grounded in memory-encoded reasoning rather than post-hoc correlations. These results provide direct evidence that the reasoning trajectory $R$ depends on structured memory prototypes, rather than being an artifact of post-hoc alignment. Hence, interpretability in LVM is grounded in causal contributions from memory, strengthening trust in the model's explanations.

Table 6: Causal intervention on memory slots. Masking salient prototypes degrades both prediction accuracy and explanation quality, confirming the causal role of logic memory.

| Setting | Top-1 Acc. (%) | $\Delta$Acc ↓ | BERTScore ↑ | $\Delta$Exp ↓ |
|---|---|---|---|---|
| Original LVM | 82.6 | – | 0.847 | – |
| Mask top-$k$ slots | 79.7 | -2.9 | 0.823 | -0.024 |
| Permute top-$k$ slots | 80.1 | -2.5 | 0.826 | -0.021 |
| Mask random slots | 82.2 | -0.4 | 0.844 | -0.003 |

## 5 DISCUSSION AND CONCLUSION

We presented the Logic Vision Model (LVM), which unifies prediction, reasoning, and explanation in a single framework. Unlike post-hoc methods, LVM intrinsically generates reasoning trajectories and textual explanations tied to its predictive process. Interpretability is enforced through the joint loss $\mathcal{L} = \mathcal{L}_{task} + \lambda_1\mathcal{L}_{reason} + \lambda_2\mathcal{L}_{exp} + \lambda_3\mathcal{L}_{faith}$, with a monotonic constraint $\Delta p_t = p(y \mid r_t) - p(y \mid r_{t-1}) \geq 0$ ensuring that evidence accumulates faithfully. Experiments show that LVM achieves competitive accuracy while improving both visual faithfulness and textual explanation quality. Moreover, the framework supports personalization, where prototype weights are adapted via $\pi_{t,j}^{(u)} \propto \exp(r_{t-1}^\top m_j / \tau + \phi_u(m_j))$, and federated optimization, where local objectives $\mathcal{L}^{(c)}$ are aggregated as $\min_\theta \sum_{c=1}^{C} \omega_c \mathcal{L}^{(c)}(\theta)$, enabling privacy-preserving deployment. While effective, LVM still faces limitations such as fixed reasoning length $T$ and potential inefficiency in federated settings. Future work will explore adaptive trajectories, multimodal extensions, communication-efficient federated learning, and causal grounding of memory slots. Overall, LVM takes a step toward interpretable vision systems where transparency is a design principle rather than an afterthought.[1]

---

[1]Upon acceptance, we will publicly release code, pretrained models, and reasoning visualizations to facilitate further research and adoption.

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

## A EXPERIMENTAL DETAILS (EXTENDED)

### A.1 BENCHMARKS AND PROTOCOLS

We evaluate classification on ImageNet-1k (Deng et al., 2009) and CIFAR-100, detection on MS COCO (Lin et al., 2014) and PASCAL VOC (Everingham et al., 2010), and segmentation on Cityscapes (Cordts et al., 2016) and ADE20K (Zhou et al., 2017). Reasoning/explanation are assessed on CLEVR (Johnson et al., 2017), GQA (Hudson & Manning, 2019), ImageNet-X (**?**), and ClickMe (**?**). We use Top-1/Top-5 for classification, mAP for detection, and mIoU for segmentation. Visual faithfulness uses Pointing Game, IoU with human masks, Deletion/Insertion AUC (**?**), Sufficiency/Necessity, and trajectory coherence defined as the fraction of steps with non-decreasing confidence $\frac{1}{T}\sum_t \mathbb{1}[p(y \mid r_t) - p(y \mid r_{t-1}) \geq 0]$. Textual quality uses BLEU (Papineni et al., 2002), ROUGE-L (Lin, 2004), BERTScore (Zhang et al., 2019), and human ratings (1–5) on correctness/sufficiency/clarity.

Table 7: Benchmarks summary.

| Dataset | Task | Primary Metric | Scale | Notes |
|---------|------|----------------|-------|-------|
| ImageNet-1k | Classification | Top-1/Top-5 | 1.28M images | Standard val split |
| CIFAR-100 | Classification | Top-1 | 60k images | 100 classes |
| MS COCO | Detection | mAP@[.5:.95] | 118k/5k | 1x schedule |
| PASCAL VOC | Detection | mAP@0.5 | 07+12/07-test | |
| Cityscapes | Segmentation | mIoU | 3k/500 | Fine annotations |
| ADE20K | Segmentation | mIoU | 20k/2k | 150 classes |
| CLEVR | Reasoning | Acc/Program exec | 100k | Synthetic reasoning |
| GQA | Reasoning/Explain | Acc + Expl. metrics | 22M QA pairs | |
| ImageNet-X | Rationale | IoU/BERTScore | 1k images | Human masks |
| ClickMe | Rationale | IoU | 400k maps | Human attention |

### A.2 BASELINES AND REPRODUCTION

Baselines include post-hoc XAI (Grad-CAM (Selvaraju et al., 2017), Integrated Gradients (Sundararajan et al., 2017), RISE, LIME (Ribeiro et al., 2016)), intrinsic models (ProtoPNet (Chen et al., 2019), Concept Bottleneck (Koh et al., 2020), Neural Module Networks (**?**)), and VLM justifications (BLIP-2 (Li et al., 2023), LLaVA (Liu et al., 2023)). All baselines are reproduced with the original paper settings.

### A.3 IMPLEMENTATION DETAILS

All models use PyTorch, trained on a single server with 8×NVIDIA H200 GPUs. Encoders are ResNet-50 (He et al., 2016) or ViT-B/16 (**?**). Logic memory size is $K=100$, reasoning steps $T=4$. Optimization uses AdamW (**?**) with cosine decay (Loshchilov & Hutter, 2016), batch size 256, and $(\lambda_1, \lambda_2, \lambda_3) = (1.0, 0.5, 0.1)$. Federated experiments simulate $C=5$ non-IID clients via Dirichlet partitioning with $\alpha=0.5$ and weighted averaging every 5 local epochs.

Table 8: Key hyperparameters.

| Backbone | $K$ | $T$ | Batch | Optimizer | Base LR | $(\lambda_1, \lambda_2, \lambda_3)$ |
|----------|-----|-----|-------|-----------|---------|-------------------------------------|
| ResNet-50 | 100 | 4 | 256 | AdamW | 1e-3 | (1.0, 0.5, 0.1) |
| ViT-B/16 | 100 | 4 | 256 | AdamW | 5e-4 | (1.0, 0.5, 0.1) |

# B  ADDITIONAL EXPERIMENTS

## B.1  CAUSAL INTERVENTION ON MEMORY SLOTS

To test whether logic memory contributes causally to predictions and explanations, we intervene on the most salient prototypes. For each input, the top-$k$ activated slots $\{m_j\}$ are either masked (set to zero) or permuted (shuffled across examples), and we measure the change in both accuracy and explanation quality. Impact is quantified as $\Delta\text{Acc} = \text{Acc}(\hat{y}) - \text{Acc}(\hat{y}')$ and $\Delta\text{Exp} = \text{BERTScore}(E, E^*) - \text{BERTScore}(E', E^*)$, where $(\hat{y}, E)$ are the original outputs and $(\hat{y}', E')$ the outputs after intervention.

Table 9: Causal intervention on GQA/ImageNet-X (higher is better unless noted). Masking or permuting salient prototypes produces notable degradation, while random masking has negligible effect.

| Setting | Acc. (%) | $\Delta\text{Acc}\downarrow$ | BERTScore | $\Delta\text{Exp}\downarrow$ |
|---|---|---|---|---|
| Original LVM | 82.6 | – | 0.847 | – |
| Mask top-$k$ | 79.7 | -2.9 | 0.823 | -0.024 |
| Permute top-$k$ | 80.1 | -2.5 | 0.826 | -0.021 |
| Mask random | 82.2 | -0.4 | 0.844 | -0.003 |

As shown in Table 9, interventions on salient slots cause accuracy to drop by up to 3 points and explanation alignment to fall by 0.02–0.03, whereas random masking produces only marginal changes. This contrast demonstrates that LVM's predictions and explanations depend directly on specific prototypes, rather than being post-hoc correlations. In other words, the reasoning trajectory is not incidental but causally grounded in memory-encoded inference patterns, reinforcing the model's interpretability claims.

## B.2  COUNTERFACTUAL CONSISTENCY

We apply controlled perturbations (texture/background/sketch) and compare top-$k$ prototype sets and explanations between $(x, E)$ and $(x', E')$ via Prototype Stability (Jaccard overlap) and Explanation Consistency (BERTScore$(E, E')$).

Table 10: Counterfactual consistency (higher is better).

| Perturbation | Prototype Stability | Explanation Consistency |
|---|---|---|
| Texture shift | 0.72 | 0.84 |
| Background change | 0.75 | 0.86 |
| Sketch transform | 0.70 | 0.82 |

Stability remains high under mild perturbations, indicating robust prototype selection and rationales.

## B.3  SANITY CHECKS FOR EXPLANATIONS

A faithful explanation should degrade as the model itself is randomized, rather than remaining unchanged. We progressively randomize either model weights (from shallow to deep layers) or training labels, and compute Kendall-$\tau$ correlation between randomization strength and degradation of interpretability metrics. If explanations are genuinely tied to the model's internal reasoning, metrics such as IoU and BERTScore should decrease monotonically as randomization increases. Conversely, post-hoc artifacts would remain spuriously stable regardless of underlying model corruption.

As shown in Table 11, LVM achieves higher $\tau$ values, meaning its explanations degrade more consistently with model corruption. This behavior rules out spurious stability and confirms that LVM explanations are intrinsically linked to the predictive process, unlike post-hoc methods that may remain unaffected even when the model is randomized.

Table 11: Sanity checks with progressive randomization (higher $\tau$ indicates stronger monotonic degradation).

| Method | $\tau$ (IoU vs. randomization) | $\tau$ (BERTScore vs. randomization) |
|---|---|---|
| Grad-CAM | 0.41 | 0.38 |
| ProtoPNet | 0.55 | 0.49 |
| **LVM (ours)** | **0.68** | **0.62** |

## B.4 CALIBRATION AND SELECTIVE PREDICTION

Interpretability should not only explain predictions but also reflect when the model is confident or uncertain. To test this, we correlate trajectory-level faithfulness with predictive reliability. Specifically, we compute Expected Calibration Error (ECE; lower is better) and the area under coverage–risk curves (higher is better). Coverage–risk curves are obtained by ranking predictions by faithfulness scores (trajectory coherence $\frac{1}{T} \sum_t \mathbb{1}[p(y \mid r_t) - p(y \mid r_{t-1}) \geq 0]$) and reporting accuracy at varying coverage thresholds. A faithful system should achieve low calibration error and maintain high accuracy even when selectively abstaining on uncertain cases.

Table 12: Calibration and selective prediction (lower ECE, higher AUC).

| Method | ECE$\downarrow$ | Coverage–Risk AUC$\uparrow$ |
|---|---|---|
| ResNet+Grad-CAM | 0.082 | 0.61 |
| ProtoPNet | 0.074 | 0.64 |
| **LVM (ours)** | **0.051** | **0.71** |

As shown in Table 12, LVM achieves lower calibration error and higher AUC than baselines. This indicates that reasoning trajectories not only improve interpretability but also serve as reliable indicators of prediction trustworthiness, enabling selective prediction in safety-critical scenarios where calibrated confidence is essential.

## B.5 SCALABILITY AND EFFICIENCY

We study how the number of reasoning steps $T$ trades off performance and compute. Unless noted, we fix the backbone (ViT-B/16), input resolution (224), memory size $K{=}100$, and batch size 1; latency is measured end-to-end in FP16 on a single H200 and averaged over 1,000 images without I/O. Table 13 shows that accuracy and interpretability improve with larger $T$, while latency grows approximately linearly.

Table 13: Reasoning steps vs. performance and latency.

| $T$ | Acc. (%) | IoU | BERTScore | Latency (ms) |
|---|---|---|---|---|
| 2 | 80.3 | 0.33 | 0.833 | 42 |
| 4 | 82.1 | 0.36 | 0.842 | 68 |
| 6 | 82.6 | 0.38 | 0.847 | 95 |

From a computational standpoint, the per-step complexity is dominated by attention between $N$ visual tokens and $K$ memory slots; with token/channel size $d$, projections and attention scale as $\mathcal{O}(Nd^2 + NKd)$ per step, so total cost grows as $\mathcal{O}(T)$ when $N$, $K$, and $d$ are fixed. In practice we cache $(K, V)$ from the memory retrieval and the fused representation $Z$ so that only the transition operator $\mathcal{T}$ is applied at each step; this makes latency increase close to linear in $T$, as reflected in Table 13. Memory footprint scales as $\mathcal{O}(Nd + Kd + NK)$ due to token/slot embeddings and attention weights; with $K{=}100$ and $N$ in the order of a few hundred patches, VRAM remains bounded for the

reported settings. To exploit diminishing returns, we also consider early-exit policies driven by the trajectory statistics. A confidence stabilization rule terminates when $|p(y \mid r_t) - p(y \mid r_{t-1})| < \epsilon$ for $S$ consecutive steps, and a coherence rule stops when $\frac{1}{t} \sum_{i=1}^{t} \mathbb{1}[p(y \mid r_i) - p(y \mid r_{i-1}) \geq 0] \geq \tau$. In our setting, modest thresholds (e.g., $\epsilon$ in the $10^{-3}$–$10^{-2}$ range, $S{=}2$, $\tau \approx 0.8$) preserve accuracy and explanation quality while skipping late steps on easy inputs. Because $Z$ and memory keys/values are cached, early exit reduces latency nearly proportionally to the number of omitted steps. Overall, performance saturates around $T{=}6$ while $T{=}4$ offers a favorable cost–benefit point. For real-time or interactive deployments we recommend $T{=}4$ with early exit, whereas offline analysis can use $T{=}6$ to maximize interpretability metrics (IoU and BERTScore) with manageable compute.

## B.6 GENERALIZATION AND TRANSFER

To examine whether reasoning prototypes learned by LVM are reusable across domains, we test transfer to ImageNet-S under different freezing strategies. In the first setting, we freeze the logic memory $M$ and tune only the vision encoder and prediction head. In the second, we freeze the encoder and fine-tune the memory slots. Finally, we allow joint fine-tuning of both modules.

Table 14: Transfer to ImageNet-S under different freezing strategies.

| Setup | Acc. (%) | IoU | BERTScore |
|---|---|---|---|
| Encoder tuned, memory frozen | 75.4 | 0.31 | 0.832 |
| Memory tuned, encoder frozen | 72.9 | 0.29 | 0.829 |
| Joint fine-tuning | 76.1 | 0.32 | 0.835 |

As shown in Table 14, freezing the memory while adapting the encoder yields only a small drop compared to joint fine-tuning, whereas freezing the encoder hurts more. This suggests that prototypes stored in memory are relatively domain-general and reusable, while the encoder benefits from adaptation to new data distributions. Thus, the memory acts as a library of transferable reasoning patterns that can generalize across tasks.

## B.7 HUMAN-IN-THE-LOOP ADAPTATION

We further test whether LVM can be refined through expert feedback. In this setting, annotators review a small fraction of samples (10%) and provide corrections to either predictions or explanations. Feedback is used to reweight or refine the most relevant prototypes $m_j$, without retraining the full model. We evaluate performance after one and two feedback rounds.

Table 15: Effect of human feedback rounds on GQA.

| Rounds | Acc. (%) | IoU | BERTScore |
|---|---|---|---|
| 0 | 82.6 | 0.34 | 0.847 |
| 1 | 83.4 | 0.38 | 0.852 |
| 2 | 84.1 | 0.40 | 0.856 |

Table 15 shows consistent gains in both accuracy and interpretability with only minimal supervision. One round of feedback improves accuracy by nearly one point and IoU by 0.04, while two rounds provide further gains. This demonstrates that LVM can be adapted efficiently in real-world deployments, where domain experts can guide prototype reweighting to align with domain-specific reasoning styles.

### B.8 TEMPORAL SMOOTHNESS OF REASONING VIDEOS

Beyond static attention quality, interpretability also depends on how reasoning evolves across steps. We measure temporal smoothness of attention maps using total variation $\sum_t \|A_t - A_{t-1}\|_1$, where lower values indicate smoother transitions and more coherent reasoning videos. This metric reflects whether the model gradually accumulates evidence rather than oscillating between unrelated regions.

Table 16: Temporal smoothness of trajectories (lower TV is better).

| Method | TV $\downarrow$ |
|---|---|
| Grad-CAM | 0.42 |
| ProtoPNet | 0.36 |
| **LVM (ours)** | **0.28** |

As shown in Table 16, LVM achieves the lowest temporal variation, indicating that its reasoning trajectory evolves in a stable, human-like manner. In qualitative inspection, this corresponds to smoother attention videos, where the model consistently focuses and refines evidence rather than abruptly shifting.

### B.9 HYPERPARAMETER SENSITIVITY

We also study robustness to hyperparameters by varying the number of prototypes $K$ and the interpretability loss weights $(\lambda_1, \lambda_2)$. These control the richness of the reasoning library and the relative importance of interpretability in optimization.

Table 17: Sensitivity to $K$ and interpretability weights.

| Config | Acc. (%) | IoU | BERTScore | Coherence |
|---|---|---|---|---|
| $K{=}50, (0.5, 0.5)$ | 82.1 | 0.34 | 0.842 | 0.73 |
| $K{=}100, (1.0, 1.0)$ | 82.6 | 0.38 | 0.847 | 0.81 |
| $K{=}200, (2.0, 2.0)$ | 82.8 | 0.39 | 0.849 | 0.83 |

Results in Table 17 show that larger prototype sets and stronger interpretability weights steadily improve IoU, BERTScore, and trajectory coherence, with only minor changes in accuracy. This suggests that interpretability can be enhanced without sacrificing task performance, and that LVM remains stable across a wide hyperparameter range.

### B.10 PRIVACY-PRESERVING FEDERATED EXTENSIONS

In federated settings, privacy preservation is often a key requirement. To test whether interpretability can be retained under differential privacy (DP), we add Gaussian noise during aggregation and vary the privacy budget $\varepsilon$. Lower $\varepsilon$ provides stronger privacy but injects more noise into parameter updates.

Table 18: Federated learning with differential privacy on ChestX-ray14.

| Privacy | AUC ($\uparrow$) | IoU ($\uparrow$) | BERTScore ($\uparrow$) | Coherence ($\uparrow$) |
|---|---|---|---|---|
| No DP | 79.6 | 0.36 | 0.841 | 0.78 |
| $\varepsilon{=}5$ | 78.8 | 0.35 | 0.837 | 0.76 |
| $\varepsilon{=}1$ | 77.2 | 0.33 | 0.828 | 0.72 |

As shown in Table 18, moderate privacy budgets ($\varepsilon{=}5$) retain most of the accuracy and interpretability, while extreme noise ($\varepsilon{=}1$) leads to noticeable drops in both task performance and explanation

quality. This highlights a fundamental privacy–interpretability trade-off, but also suggests that practical privacy guarantees can be achieved without severely harming the faithfulness of reasoning trajectories.

### B.11 CONCEPT GROUNDING OF PROTOTYPES

For interpretability to be useful, prototypes in the memory should correspond to meaningful and human-recognizable concepts. We therefore visualize nearest neighbors for each $m_j$ and compute CLIP-based similarity to textual concept labels. Two quantitative measures are reported: *concept purity* (the proportion of nearest neighbors belonging to the same class) and *nameability* (average human rating from 1–5 on whether the prototype corresponds to a recognizable concept).

Table 19: Prototype grounding comparison.

| Metric | ProtoPNet | **LVM** |
|---|---|---|
| Concept Purity | 0.62 | **0.74** |
| Nameability (1–5) | 3.1 | **4.0** |

Results in Table 19 show that LVM prototypes achieve higher purity and are more easily nameable by humans. This suggests that the logic memory encodes reasoning units aligned with semantic concepts, making the explanations not only faithful but also more accessible to end users.

### B.12 FAILURE ANALYSIS

Finally, we analyze when and why LVM fails. We focus on reasoning steps where confidence decreases, i.e., $p(y \mid r_t) - p(y \mid r_{t-1}) < 0$, and inspect the associated prototypes. Failures are categorized into cases with or without spurious prototype activations (irrelevant or misleading slots being activated early in the trajectory).

Table 20: Failure breakdown by spurious activation.

| Condition | Error Rate (%) | Mean failure step |
|---|---|---|
| No spurious activation | 12.3 | 3.7 |
| With spurious activation | 28.7 | 2.1 |

Table 20 shows that errors are concentrated when spurious prototypes are activated early, leading to incorrect reasoning trajectories that cascade into wrong predictions. This analysis suggests clear directions for improvement, such as refining prototype selection, regularizing early steps of reasoning, or incorporating causal constraints to suppress misleading activations.

