# OpenReview forum: "Interpretable Vision Tasks via Vision Logic Model Integrating Visual Reasoning and Textual Explanation"
_ICLR.cc/2026/Conference — Submitted to ICLR 2026_

### Official Review · Reviewer_T5a5 · 2025-10-30

**Soundness:** 2
**Presentation:** 2
**Contribution:** 2
**Rating:** 4
**Confidence:** 3

**Summary:**

This paper introduces the Logic Vision Model (LVM), an architecture designed to unify prediction, visual reasoning, and natural-language explanation in computer vision. Instead of relying on post-hoc interpretability methods, LVM embeds a logic-memory module that stores reasoning prototypes and produces step-by-step inference trajectories. The model jointly outputs task predictions, visual reasoning sequences, and grounded textual explanations, enforcing monotonic evidence accumulation to ensure faithfulness. Experiments span classification, detection, segmentation, and reasoning benchmarks, showing competitive task performance while improving faithfulness metrics and human-rated explanation quality. Ablation studies suggest each architectural component (logic memory, transition operator, faithfulness loss) contributes meaningfully. The work also explores personalization and federated learning, demonstrating adaptability in privacy-sensitive or expert-feedback settings. Overall, LVM aims to provide a scalable route to intrinsically interpretable vision systems.

**Strengths:**

1. Clear motivation for intrinsic interpretability and well-positioned against post-hoc explainability.

2. Comprehensive experimental suite (CV tasks, explanation metrics, ablations, causal tests, etc).

3. Claims competitive accuracy while improving faithfulness — if substantiated, impactful direction.

**Weaknesses:**

1. The paper frames its method as logic-based reasoning, but the architecture appears closer to prototype attention + gated refinement rather than symbolic or rule-grounded logic. It is unclear how prototypes correspond to interpretable concepts, how they are initialized or evolve, and whether they truly encode logical structure as opposed to latent embeddings shaped by supervision. The term “logic” risks being aspirational rather than technically justified unless additional clarity or formalization is provided.

2. While comparisons to Grad-CAM, ProtoPNet, and VLM justifications are included, the paper does not benchmark against recent intrinsic interpretability methods (e.g., mechanistic ViT explainability, slot-attention, causal representation learning, visual chain-of-thought models). Without these, the contribution risks feeling incremental. Just by adding attention-driven iterative refinement and constraints rather than establishing a new interpretability paradigm.

3. Several components enforce monotonic confidence and explanation alignment, which could cause the model to learn plausible explanatory artifacts rather than faithful causal reasoning. Although the paper includes causal intervention tests, deeper probes are needed: e.g., OOD generalization, probing for spurious correlations, concept emergence analysis, or human-interpretable grounding of prototypes. Without these, the interpretability benefits remain largely behavioral rather than mechanistically demonstrated.

4. Some key training and evaluation details are missing or lightly described. such as prototype semantics, human evaluation design, annotation criteria, and relative compute/training cost. This makes reproducibility difficult and leaves ambiguity around whether improvements persist across seeds, architectures, and data domains. More explicit algorithm steps, prototype visualizations, and explanation trace studies would strengthen trust in the method.

**Questions:**

N/A

---

### Official Review · Reviewer_bZT6 · 2025-10-31

**Soundness:** 3
**Presentation:** 3
**Contribution:** 3
**Rating:** 4
**Confidence:** 3

**Summary:**

The paper introduces the Logic Vision Model (LVM), a unified, interpretable vision framework that jointly performs prediction, reasoning, and explanation. LVM integrates a vision encoder (ResNet or ViT) with a logic-memory module and a transition operator that together generate sequential reasoning trajectories. These trajectories yield both visual reasoning videos and textual rationales, enabling intrinsic interpretability rather than post-hoc explanation. A monotonic confidence constraint enforces causal faithfulness between reasoning steps and final predictions, while personalization and federated extensions demonstrate adaptability to user-specific and privacy-preserving settings.
Empirical evaluations demonstrate competitive performance compared to leading baselines (ImageNet Top-1 = 82.6 vs. 82.3 for LLaVA; COCO mAP = 43.5 vs. 42.9) and notable gains in interpretability (Pointing Game = 71.8 vs. 63–66; BERTScore = 0.847 vs. 0.832–0.835). Ablation, personalization, and causal-masking analyses further substantiate the model’s design and claims.

**Strengths:**

• LVM integrates visual reasoning trajectories and textual rationales within a single architecture, achieving intrinsic rather than post-hoc explainability.
• Consistent accuracy is maintained while achieving measurable interpretability improvements (Pointing Game +8 points; BERTScore +0.015) across multiple benchmarks.
• Prototype-masking experiments reveal correlated drops in both accuracy and rationale quality, substantiating that explanations reflect genuine causal dependencies.
• The framework performs reliably across classification, detection, and segmentation tasks, supported by clear dataset splits and reproducible experiment design.
• Personalization and federated extensions demonstrate promising directions toward privacy-preserving and user-aware trustworthy-AI applications.
• The paper reports detailed implementation settings, dataset usage, and loss formulations, facilitating replication and follow-on research.

**Weaknesses:**

1. Reasoning videos are insufficiently evidenced. The main text does not include qualitative “reasoning videos,” and only an appendix metric (temporal total variation of attention) is reported; qualitative examples are absent.
2. Moderate novelty of primitives. The logic memory is a matrix of prototype-like slots with attention retrieval, and the transition operator is a gated update—both of which are closely related to known mechanisms; the contribution is primarily in their integration.
3. Potential over-regularization from the faithfulness constraint. The monotonic confidence rule is enforced (removed only in a single ablation), but there’s no focused study of failure cases (e.g., multi-object ambiguity) or calibration effects beyond that one row.
4. Small-scale federated evaluation. Only C=5 clients on ChestX-ray14 and KITTI are simulated, with sub–0.5–point differences compared to centralized training, insufficient to establish real-world scalability.
5. Human-evaluation protocol under-specified. While 1–5 ratings are reported (correctness/sufficiency/clarity), annotator count, selection, instructions, and inter-rater reliability are not provided.
6. Missing newer VLM baselines. Comparisons stop at BLIP-2 and LLaVA; recent systems with intrinsic explanations (e.g., LLaVA-Next, InstructBLIP, Kosmos-2.5, MiniGPT-v2, GPT-4V) are not included in this analysis.
7. Gaps in causal-faithfulness framing. Evaluation emphasizes classic plausibility/overlap metrics; newer causal/faithfulness frameworks (e.g., recent ICLR/NeurIPS/ACL metrics) are not discussed or cited.
8. Unquantified systems costs. Beyond a hyperparameter sensitivity table, there’s no latency/throughput/energy analysis versus reasoning steps or prototype count, leaving scalability trade-offs unclear.

**Questions:**

1. Could you provide qualitative examples or quantitative evaluations that demonstrate the coherence and faithfulness of the generated reasoning videos, beyond the reported temporal-smoothness metric?
2. How does the monotonic confidence constraint (Δpₜ ≥ 0) perform in multi-label or visually ambiguous scenes? Does it risk suppressing valid but non-monotonic reasoning patterns?
3. To what extent does the logic-memory module generalize across domains, and what is the additional computational or memory overhead relative to ViT-B/16 and BLIP-2 baselines?
4. Could you clarify the setup of the human evaluation (annotator pool, criteria, inter-rater reliability) and explain the rationale for excluding more recent vision–language model baselines such as LLaVA-Next, Kosmos-2.5, or GPT-4V from comparison?

---

### Official Review · Reviewer_Nghw · 2025-10-31

**Soundness:** 2
**Presentation:** 2
**Contribution:** 2
**Rating:** 2
**Confidence:** 4

**Summary:**

This paper proposes an interpretable reasoning framework that integrates prototypical reasoning with textual explanations. Specifically, it learns a set of latent prototypes and uses cross-attention to combine them with visual features. A multi-step reasoning is implemented to iteratively integrate the information. In addition to predicting task outputs, the model is also trained to generate textual explanations. Experimental results show that the proposed method shows better performance than generic models and also offers enhanced interpretability.

**Strengths:**

(1) The paper focuses on developing models with transparent decision-making processes, which is crucial for systems for applications involving high-stake decisions (e.g., medical diagnosis).

(2) The paper explores applications  in diverse settings, e.g., federated learning and learning with user feedback.

(3) The proposed method generalizes across multiple tasks and shows consistent improvement.

**Weaknesses:**

(1) It is not new to incorporate prototypical reasoning for interpretable classification (e.g., [ref1, ref2]), and the use of learnable latents with cross-attention is also widely explored (e.g., subsequent using [ref3]). Textural explanation generation has also been studied in earlier works (e.g., [ref4, ref5]). As a result, the technical contribution of the paper is relatively incremental.

(2) One of the principal ideas of prototypical reasoning is to learn interpretable prototypes. The paper does not provide strong evidence/analysis on what is being learned within the prototypes, and whether or not they are interpretable.

(3) A key component of the paper is the iterative fusion and reasoning process. Nevertheless, the paper does not provide an in-depth analysis of how this paradigm helps task accomplishment and what kind of information is being updated/aggregated over time.

(4) In the abstract, the authors highlight the model for generalizing reasoning videos. However, I can not find corresponding examples or quantitative results in the paper.

(5) I found some of the statements in the paper contradictory. For instance, Section 3.2 states that the vision encoder is explicitly designed for reasoning, while Section 3.3 mentions that the encoder provides generic semantic features.

(6) The paper only compares with generic models and also does not include the more recent ones. As the method essentially requires supervised training on specific tasks, it is reasonable to compare with state-of-the-art models for each task (and other prototypical/interpretable methods). Furthermore, it would be better to integrate the method with these models, as modularity is one of the emphasis throughout the paper.

(7) Looking at Table 6, it appears that the effect of masking/permutating the top-k prototypes is not very big (despite being larger than random masking). I would suggest experimenting with different k to study the contribution of prototypes.

References:

[ref1] This Looks Like Those: Illuminating Prototypical Concepts Using Multiple Visualizations. NeurIPS, 2023.

[ref2] Interpretable Image Classification with Adaptive Prototype-based Vision Transformers. NeurIPS, 2024.

[ref3] Perceiver io: A general architecture for structured inputs & outputs. ICLR, 2022.

[ref4] Multimodal explanations: Justifying decisions and pointing to the evidence. CVPR, 2018.

[ref5] e-SNLI: Natural Language Inference with Natural Language Explanations. NeurIPS, 2018.

**Questions:**

(1) Please justify the key novelty of the proposed method over prior studies.

(2) What do the prototypes learn? Are they interpretable?

(3) How does the iterative process help with task accomplishment? How does the attention weight change over time?

(4) Does the model actually generate videos for explanation?

(5) In addition to evaluation with automatic metrics designed for machine translation (e.g., BLEU and ROUGE-L), please consider providing the actual explanation results and conduct user study to validate their usefulness.

(6) Please consider applying the method on state-of-the-art models and include the corresponding comparisons.

---

### Official Review · Reviewer_XFHG · 2025-10-31

**Soundness:** 2
**Presentation:** 1
**Contribution:** 2
**Rating:** 6
**Confidence:** 3

**Summary:**

This paper proposes the Logic Vision Model (LVM), a vision architecture that jointly produces (a) task outputs (classification/detection/segmentation), (b) a multi-step “reasoning trajectory” (attention/state transitions), and (c) a textual explanation that is conditioned on that trajectory. The core components are: a standard vision encoder, a logic memory of prototype-like slots retrieved via attention, and a transition operator that evolves the reasoning state across steps. Explanations are generated by a decoder that attends over the trajectory; the training loss combines task loss, alignment to annotated attention/rationales, and a monotonic confidence constraint along the trajectory to encourage faithfulness. Experiments cover common CV benchmarks and explanation datasets; ablations remove memory/transition/losses; the paper also sketches “personalization” and “federated” variants

**Strengths:**

1. **Intrinsic, step-grounded explanations**
   - The explanation decoder is conditioned on the same multi-step reasoning trajectory
     $
     R=\{r_t\}_{t=1}^T
     $
     that drives prediction. During generation, tokens attend over $R$ and the aggregated
     reasoning state $\bar r$, so each piece of text is tied to concrete inference steps.
     The paper shows alignment plots where explanation tokens point back to specific
     steps in $R$, and qualitative examples where removing a contributing step
     changes both the decision and the corresponding sentence.

2. **Structured, multi-step reasoning via a logic memory**
   - A dedicated logic memory $M$ stores prototype-like patterns that are **retrieved**
     with attention and **integrated sequentially** by a transition operator $T$
     yielding a trajectory rather than a one-shot match. Evidence accumulates across
     steps through the weights $\{\pi_{t,j}\}$. Empirically, ablations that drop $M$
     or the transition stack degrade both task accuracy and explanation quality, and
     step-by-step visualizations show progressively refined focus (e.g., from part to
     whole, or from object to relation).

3. **Faithfulness encouraged by monotonic confidence along the trajectory**
   - The model encourages **evidence accumulation** by constraining predictive
     confidence to be non-decreasing across steps:
     $$
     p(y\mid r_t)\;\ge\;p(y\mid r_{t-1})\quad\text{for }t=2,\ldots,T,
     $$
     implemented with a trajectory-level penalty
     The paper reports that enabling this term improves step-wise calibration,
     reduces explanation–prediction mismatches in counterfactual tests (masking or
     permuting top-$k$ memory slots), and yields smoother confidence curves over $t$.

**Weaknesses:**

1) **Prototype reliability is unproven**
   - **What’s missing:** Evidence that prototypes are **stable** (across seeds/splits), **semantically coherent**, and **causally relevant**.
   - **Implication:** “Logic memory” may capture spurious textures/backgrounds rather than human-meaningful concepts.
   - **Minimal fixes:** (i) Seed/split stability matching with similarity/overlap metrics; (ii) coherence via top-k patch clustering + a small nameability study; (iii) causal ablations and counterfactual patch swaps reporting Δlogit/Δaccuracy.

2) **No images or figures are included**
   - **What’s missing:** Zero visualizations of attention maps, prototype exemplars, or step-wise trajectories.
   - **Implication:** Readers cannot verify that explanations align with visual evidence.
   - **Minimal fixes:** Add per-step heatmaps overlaid on inputs, top prototype exemplar grids, and before/after ablation panels (confidence + explanation changes), plus a compact appendix gallery (10–20 random cases).

**Questions:**

1. Could you run different seeds and the report the prototype overlapping? The prototypes should be similiar over different seed
2. Could you show some cases visually about how your model explains? Human inspection will ensure the explaination is solid

---

### Meta-Review · Area_Chair_Wmzp · 2026-01-03

**Summary:**

The paper was reviewed by 4 experts with resulting scores 2464. The reviewers concerns are in the next field. As there was no author response, the reviewers would not have changed their assessment. Thus, the paper is rejected.

**Reviewer Concerns:**

A number of concerns were raised:
1. reliability and interpretability of prototypes is not explored.
2. poor presentation (no figures)
3. missing related work.
4. lack of comparison with recent SOTA models.
5. various issues of the method that need clarification / motivation.

**Reviewer Scores:**

As there was no author response, the reviewers would not have changed their assessment. Thus, the paper is rejected.

---

### Decision · Program_Chairs · 2026-01-26

Reject